# Mathematical measures of societal polarisation

**Johnathan A. Adams**[1], **Gentry White**[1,2], **Robyn P. Araujo**[1,3]*

**1** School of Mathematical Sciences, Queensland University of Technology, Brisbane, Queensland, Australia, **2** QUT Centre for Data Science, Queensland University of Technology, Brisbane, Australia, **3** Institute of Health and Biomedical Innovation, Kelvin Grove, Queensland, Australia

* r.araujo@qut.edu.au

## Abstract

In opinion dynamics, as in general usage, polarisation is subjective. To understand polarisation, we need to develop more precise methods to measure the agreement in society. This paper presents four mathematical measures of polarisation derived from graph and network representations of societies and information-theoretic divergences or distance metrics. Two of the methods, min-max flow and spectral radius, rely on graph theory and define polarisation in terms of the structural characteristics of networks. The other two methods represent opinions as probability density functions and use the Kullback–Leibler divergence and the Hellinger distance as polarisation measures. We present a series of opinion dynamics simulations from two common models to test the effectiveness of the methods. Results show that the four measures provide insight into the different aspects of polarisation and allow real-time monitoring of social networks for indicators of polarisation. The three measures, the spectral radius, Kullback–Leibler divergence and Hellinger distance, smoothly delineated between different amounts of polarisation, i.e. how many cluster there were in the simulation, while also measuring with more granularity how close simulations were to consensus. Min-max flow failed to accomplish such nuance.

## Introduction

Polarisation occurs in society when its members fail to agree on a topic or opinion. In the simplest case, polarisation is a bifurcation of a society into two sub-groups holding disjoint opinions on a given topic. Multi-modal polarisation, or plurality, occurs when individuals cluster their opinions around more than two poles or opinion loci. The opinion dynamics literature defines polarisation and consensus as two mutually exclusive states [1], in contrast to how we describe societies in terms of *degrees* of polarisation or consensus on a continuum. Part of the reason for viewing polarisation in the opinion dynamics literature as discrete states is because the models used to describe societies and individual interactions are deterministic and view opinion points in space. This approach facilitates computation and analysis but doesn't accurately reflect the subtle variations in how individuals conceive of their opinions. The Martins model [2] conceptualises an individual's opinion as Gaussian density functions and naturally produces nuanced states of polarisation or consensus. Opinions distributed as functions on a

**Data Availability Statement:** All relevant data are within the paper and its Supporting information files.

**Funding:** Robyn P. Araujo is the recipient of an Australian Research Council (ARC) Future Fellowship (project number FT190100645) funded

by the Australian Government. https://www.arc.gov.au/ The funders had no role in study design, data collection and analysis, decision to publish, or preparation of the manuscript.

continuum create the need for more sophisticated measures of polarisation or consensus than classification into one of two broad and possibly minimally informative categories. Such measures are the subject of this paper and will provide tools to explore how models, and ultimately real-world societies, fall into polarisation or reach consensus.

We introduce, in this paper, four methods to measure polarisation in a group of individuals. Two of these methods are derived from graph theory and use graph structure to determine the agreement in a simulation. Specifically, these methods examine the graphs representing information flow between agents, i.e. how much of an opinion can be transmitted between agents. The first is the "min-max" information flow between individuals, a continuous version of the $k$-edge connectedness of a graph. The second is to calculate the spectral radius of the graph. Finally, the last two methods are based on probability theory using the mean Kullbeck-Liebler (K-L) divergence and Hellinger distance between all individuals. We will compare these four methods over a series of simulations using two models for agent interaction with various initial conditions and parameter values. We discuss how effective these methods are at measuring the polarisation of the simulations and conclude with which one we think is the best in the most general circumstance.

## Methods of measuring polarisation from the literature

One of the most commmon methods used in the literature [3–7] to measure polarisation in an agent-based simulation model is to count opinion clusters once the simulation reaches a steady state. Counting opinion clusters is a simple method, reflecting the tendency to categorise a distribution of opinions as plurality, polarisation or consensus. Early works of [3, 4] manually identified and counted opinion clusters, which can result in subjective and possibly biased results. Supervised and unsupervised clustering algorithms are available but present several technical challenges, and results can vary depending on the algorithm used. These challenges are exacerbated in more complex simulations with opinions (and clusters) represented as density functions.

We illustrate the fundamental problem of relying solely on cluster counts to measure polarisation or consensus for models representing opinion as a density function on a continuum using two cases. First, consider the situation where a simulation in a steady-state produces $n$ opinion clusters with 85% of the agents in a single cluster. Even though there are $n$ clusters (where $n$ could potentially be large), indicating polarisation, the fact that 85% of agents are in a single cluster suggests consensus. Second, consider the case where there are multiple opinion clusters with approximately equal memberships, but the clusters are diffuse, and there is considerable overlap in the 'distinct' opinions across clusters. This case also suggests consensus. Cluster counts do not tell a complete story for a model expressing opinions as density functions on a continuum.

A more advanced method of determining polarisation is [8] which is used in the more modern work of [9]. This method measures polarisation accounting for the dispersion of agents across opinion space. Specifically, if $p_+$ and $p_-$ are the proportion of agents that finished the simulation in the upper and lower half of the opinion space, respectively, then [8] defines

$$y = p_+^2 + p_-^2, \tag{1}$$

as a measure of polarisation. Agents have reached a consensus when $y = 0$ or $y = 1$, meaning that the simulation has converged either to the centre $y = 0$ or one extreme opinion space $y = 1$. When $y = 0.5$, an equal number of agents are at each extreme, indicating polarisation.

While Eq 1 contains more information about polarisation, it is limited. Eq 1 arbitrarily divides opinion space into two halves. Furthermore, the method does not distinguish between

a densely packed consensus and a more diffuse consensus. A diffuse consensus in the upper half of the opinion space will give an equivalent value of *γ* to a denser-packed consensus. This lack of distinction is a problem as it is common for later models like [2] to create a very diffuse consensus.

**Modern measures of polarisation.** A more recent approach to measuring polarisation comes from [10]. Their focus was on quantifying polarisation through summary measures. They evaluated the effectiveness of various statistical measures, from the naive variance of opinions to their novel contribution to a measure that uses kurtosis and skew. Although relevant, we seek to differ from [10] by presenting measures that have a closer connection to polarisation and are more physically meaningful.

Another novel approach originates from [11], which measures polarisation through methods established in [12]. The principle behind the measures is based on a random walk probability to travel from one cluster to another, with one method measuring the probabilities of random walks reaching one opinion cluster from another and the other calculating a 'lower dimensional distance' between all agents and calculating the average inter and intra distance between clusters. With these measures [11] was able to more rigorously demonstrate that individuals in different opinion groups tended to consume different online media and used distinct hashtags. The method we present in this paper has the same potential to more distinctly identify the impacts of polarisation on society because our methods are continuous in a similar way as these probability measures.

A particularly novel approach to quantifying polarisation is to use machine learning algorithms [13]. First communities were identified in an online space [13] used a word-context learning algorithm which develops a vector of association between a particular context and all words. In the case of an online community, the 'contexts' are individual users, and the 'words' were the message boards the individuals could post on, so the more an individual posted to a message board, the more they became associated with the message board. With the communities identified [13] sought to establish the social dimensions, the communities would lie upon and accomplished this though through a genetic learning algorithm which the authors seeded with the 'Conservative' and 'Democrat' message boards with these two message boards defining the ends of the social dimensions. With this social dimension, [13] was able to determine how polarised communities were in the online space.

## The graph theoretic approaches to measuring polarisation

Social network analysis uses graphs and graph theory to represent and explore social structures and connections between individuals. Opinion dynamics is the study of how individuals connect and influence each other's opinions. We can conceive of any opinion dynamics model as producing graphs representing the social connections and interactions between a group of individuals. Thus it is sensible to approach measuring polarisation using graph theory. A consequence of polarisation is the changes to the graph's structure representing a group's interactions, e.g. the isolation of groups of people with different opinions. As a result, the social networks of polarised compared with non-polarised groups have a starkly different graph structure. We can then use graph theory concepts, such as edge connectivity and spectral analysis, to quantify the difference in these networks, thereby quantifying the polarisation occurring. See S1 Appendix in S1 File for a review of the graph theory concepts.

**Polarisation methods based on *k*-edge-connectivity.** One of the prevalent ideas discussed in the graph structure literature [14] is the concept of connectivity or the degree of connections between individuals represented as nodes connected via edges. The simplest connectivity measure is graph density, defined as the number of edges in a graph compared to the

maximum possible number of edges. Density, while convenient and scale-free, only gives the general local connectivity of a graph while neglecting the graph's global features, such as when a graph is partitioned into two or more components, i.e. when a society divides into two or more groups of individuals. The society would be polarised in that case, but if in those groups individuals have many edges between them, measures of density will rate the graph and society as highly connected.

Component connectivity is complimentary to density. We can derive a measure of component connectivity by noting intuitively that a graph with fewer components has greater "connectivity". Consider a graph or social network $G$ consisting of $n$ individuals represented as nodes on the graph, if $G$ consists of $K(G)$ components then $(n - K(G))/(n - 1)$ is a measure of component connectivity. As $K(G) \to 1$ then the components connectivity approaches 1 and as $K(G) \to n$ the component connectivity approaches 0. In terms of polarisation, as $K(G) \to n$ individuals are forming more and more disconnected sub-groups, i.e. "bubbles" in the current colloquialism, and as $K(G) \to 1$ individuals form fewer components, i.e. are less polarised or more in consensus. Due to its focus on global characteristics, the proposed component connectivity measure does not consider the internal connectivity of components, so a weakly connected graph of size $n$ and a strongly connected graph of size $n$ can have the same measure of component connectivity [14]. A more refined version of this component connectivity metric is in [15] but suffers from the same issue of evaluating weakly connected graphs as equivalent to strongly connected graphs [14]. In terms of polarisation, component connectivity only measures the degree that society has partitioned itself into distinct components, not the measure of a component's internal cohesion or communication.

A more compelling measure of graph connectivity originates from the concept of cutsets. Consider a graph $G$ consisting of $V$ vertices or nodes and $E$ edges connecting the nodes, $G = [V, E]$, a cutset is a subset of the edges $H \subset E$ or nodes $H \subset V$ such that if we remove $H$, the number of disconnected components in $G$ increases. When $H$ is a subset of nodes, it is known as a vertex cut. Likewise, when $H$ contains edges, it is known as an edge cut [14]. The minimum cutset of $G$ is the cutset of either edges or nodes with the smallest size. The larger a graph's minimum cutset is, the more "connected" the graph. Therefore, we can measure a graph's connectivity by finding $k$ the size of the minimum cutset. The size of the minimum edge cutset is the $k$-edge-connectedness of a graph, and likewise, the $k$-vertex-connectedness is the size of the minimum vertex cutset [14]. In social network models, we assume that $V$, the nodes or individuals, are fixed, whereas edges are the connections between individuals, and connections are a direct measure of a society's divisiveness. For this reason, we consider $k$-edge connectivity a more reasonable and intuitive measure of polarisation.

**Spectral analysis.** One important concept about graphs is their adjacency matrix representations. An adjacency matrix $A$ of a graph $G$ has elements such that $A_{ij} = 1$ if edge $e_{ij}$ exists between agents $i$ and $j$, and $A_{ij} = 0$ when $e_{ij}$ does not exists. If $G$ is a weighted graph then $A_{ij} = f(e_{ij})$ where $f$ the weighting function. We can then investigate network properties using spectral analysis of a graphs adjacency matrix. Spectral analysis uses the eigenvalue decomposition of matrices to summarise and identify characteristics of the network. The spectral radius of a matrix is an important part of spectral analysis. Denoted by $\rho(A)$, the spectral radius of a matrix $A$ is its largest eigenvalue in magnitude. What is important about the spectral radius is its relationship to the connectivity of a graph. We can illustrate this relationship by considering a society that has polarised into $m$ distinct opinion clusters. In a modelling sense this happens when agents share information completely (100%) inside a cluster and no information (0%) outside the cluster. We can represent this as a graph that is composed of $m$ complete subgraphs (i.e. fully connected subgraphs). We can then express this graph as an adjacency matrix such

that

$$A = \begin{bmatrix} J_{n_1} & 0 & \dots & 0 \\ 0 & J_{n_2} & \dots & 0 \\ \vdots & \vdots & \ddots & \vdots \\ 0 & 0 & \dots & J_{n_m} \end{bmatrix},$$

where $J_{n_i}$ is the $n_i \times n_i$ matrix of ones and $n_i$ is the size of the $i$th opinion cluster. From Theorem 1 in S2 Appendix in S1 File the spectral radius of $A$ is the size of the largest opinion cluster.

It is clear how the largest opinion cluster's size relates to polarisation. If the size of the largest cluster is the total number of individuals in society, then that society is in consensus. So we can then use the fraction of individuals in the largest cluster to measure how close society is to consensus. Theorem 1 shows that the spectral radius is the largest cluster size when a society is divided distinctly into opinion clusters. The main advantage of the spectral radius is that we can calculate the spectral radius even when opinion clusters aren't distinct and when there is a significant overlap between clusters. So the spectral radius offers us a method to estimate the size of the 'largest cluster,' which allows us to use, more broadly, the largest cluster size as a measure for polarisation.

## The information theoretic approaches to measuring polarisation

Some opinion dynamics models [2, 16] views opinions as probability distributions so it follows to use $f$-divergences as ways to quantify differences between two agents' opinions [17, 18]. The $f$-divergences measure distance between two probabilistic objects and are a 'statistical distance'. For models that consider agent opinions as probability distributions, $f$-divergences like the Kullback–Leibler divergence and the Hellinger distance can provide insights into polarisation. As for models that don't consider agent opinions as probability distributions, like HK bounded confidence, we can interpret agent opinions in a probabilistic way.

**Kullback–Leibler divergence.**  Kullback–Leibler divergence (KLD) is a measure of the difference between two probability distributions [19]. The literature uses KLD to compare models of statistical inference for Bayesian statistics. The continuous version of the K–L divergence is

$$\text{KLD}(f||g) = \int_{-\infty}^{\infty} f(x) \ \log \ \left( \frac{f(x)}{g(x)} \right) \ dx, \qquad (2)$$

where $f$ and $g$ are the probability density functions [19]. Note that $\text{KLD}(f||g) \neq \text{KLD}(g||f)$. The principle is to maximise the KLD of the posterior and prior distributions, which is equivalent to maximising over the likelihood in Bayesian statistics [20, 21]. Because of KLD's link to the likelihood in Bayesian statistics, it is sensible to use KLD as a measure of distance between agent's opinions in the Martins model [2] due to the model's reliance on Bayesian inference for generating polarisation. Taking the KLD between two agents' opinions would be treating one agent's opinion as a theoretical prior and the other's opinion as a theoretical posterior, and KLD would then reveal how much information is required for the prior agent to adopt the posterior agent's opinion. Therefore finding the KLD between all agents in a simulation to find the mean of all the inter-agent KLDs, i.e. the mean KLD, should reveal how 'close' agents are in opinion, thereby revealing how polarised the simulated society is.

**Hellinger distance.**  Similar to KLD the Hellinger distance is the distance between two probability density functions $f$ and $g$ [22–24], except Hellinger distance qualifies as a distance

metric [22] whereas KLD does not. The basis of the Hellinger distance is the Hellinger affinity [25] defined as

$$\int_X \sqrt{f(x)g(x)} \ \mathrm{d}x.$$

When $f(x) = g(x) \ \forall x \in X$ the Hellinger affinity is 1 thus the squared Hellinger distance is

$$H^2(f,g) = 1 - \int_X \sqrt{f(x)g(x)} \ \mathrm{d}x \tag{3}$$

[22–24]. We can interpret the Hellinger distance as the analogue of Euclidean distance from space vector but for probability distributions. Hence it follows to calculate the Hellinger distance between agents in the Martins model like we have suggested with the KLD and like with KLD we can find the mean Hellinger distance between every agent pair in a simulation to measure the polarisation of the simulation. Due to the Hellinger distance being a distance metric, the Hellinger distance has several advantages over KLD, chief of which is the Hellinger distance's symmetry, i.e. $H(f, g) = H(g, f)$ which halves the number of computations when calculating the mean Hellinger distance.

## Methods

In this section, we shall discuss how the methods we used to determine the effectiveness of each measure of polarisation. We developed two types of simulations using two distinct models of agent interaction. In both types, we varied core parameters which influenced polarisation in the selected models. We then applied the four measures of polarisation to every simulation. Since these measures rely on the structural elements of the social network to measure polarisation, we shall have agents interact in an open 'everyone can talk to every' environment to not bias toward polarisation.

### Agent interaction models for the simulations

We used two interaction models in the simulations for this paper: the Martins model [2, 16, 26] and the Hegselmann-Krause (H-K) Bounded Confidence model [3]. The Martins model is the newest and generates complex behaviour in simulations. The H-K Bounded Confidence model is older than the Martins model, but the behaviour it produces in simulations is well understood.

The model first proposed in [2, 16], which we shall call the Martins model, is a simple updating rule for agents derived from Bayesian inference. Because the model operates in a Bayesian framework, each agent's opinion is a guess at a true value. An agent's opinion follows a normal distribution where $x$ is the mean of that normal distribution, and $\sigma$ is the standard deviation. The value $x$ is the location of that agent's opinion, and the standard deviation $\sigma$ is an agent's uncertainty in their opinion, their strength of belief.

Eqs 4 and 5 describes the how agents update their opinion. Martins creates polarisation through the parameter $p$, where $p$ is the probability an agent shares useful information with another to update their opinion. Essentially $p$ is a global trust rate. The effect of including $p$ is that when two agents interact, their opinions are updated using $0 < p^* < 1$, which measures how much the two agents trust each other. $p^*$ is affected by how distant their opinions are

relative to their uncertainties [26].

$$x_i(t+1) = p^* x_i(t) + (1-p^*) \frac{x_i(t)/\sigma_i(t) + x_j(t)/\sigma_j(t)}{1/\sigma_i(t) + 1/\sigma_j(t)}, \tag{4}$$

$$\sigma_i^2(t+1) = \left(1 - \frac{\sigma_i^2(t)}{\sigma_j^2(t) + \sigma_i^2(t)}\right)\sigma_i^2(t) + p^*(1-p^*)\left(\frac{x_i(t) - x_j(t)}{1 + \sigma_j^2(t)/\sigma_i^2(t)}\right)^2, \tag{5}$$

where

$$p^* = \frac{p\phi(x_i(t) - x_j(t), \sqrt{\sigma_i^2(t) + \sigma_j^2(t)})}{p\phi(x_i(t) - x_j(t), \sqrt{\sigma_i^2(t) + \sigma_j^2(t)}) + 1 - p},$$

and

$$\phi(\mu, \sigma) = \frac{1}{\sigma\sqrt{2\pi}} e^{\frac{-\mu^2}{2\sigma^2}}$$

The Martins model, along with it's extension in [26], is a compelling explanation of polarisation with $p$ and $p^*$. It also produces novel behaviour. A good measure of polarisation might explain the model's behaviour. In [2, 16] the model used an unshared uncertainty assumption, where an agent could not share their $\sigma$, we use a variant of the model with that assumption relaxed [26].

The Hegselmann-Krause (H-K) Bounded Confidence model [3] is a well-studied model. It was one of the first models to create polarisation reliably. Each agent has a continuous opinion $x$. An agent $i$ will update their opinion by first taking all agents' opinions in the interval $[x_i - \epsilon, x_i + \epsilon]$, where $x_i$ is the opinion of agent $i$, and $\epsilon$ is a parameter set by the model. Agent $i$'s new opinion is the mean of all the opinions in the interval. The H-K model will serve to calibrate the new polarisation metric.

## Applying the graph theory measures

For the graph-theoretic measures to work appropriately, we need to establish graphs of the interpersonal connection between agents in a simulation. Specifically, we need the induced adjacency matrix of that graph representing the information flow between agents. The Martins model has a built-in measure of an agent's ability to compromise with other agents, $p^*$. Practically $p^*$ represents how much an agent accepts the opinion of another agent, i.e. influenced by the agent. Thus, an adjacency matrix created from $p^*$ represents the information flow of all agents in a simulation.

The H-K Bounded Confidence model has a more intuitive adjacency matrix with entries equal to 1 or 0. If agent $i$ and $j$'s opinions are within $\epsilon$, then the of the $i$th row and $j$th column in the adjacency matrix is 1 because agents within $\epsilon$ of each others' opinion will have maximum influence on each other due to the updating rules of the H-K Bounded Confidence model. For the same reason, the $i$th row and $j$th column in the adjacency matrix will be 0 when agent $i$ and $j$'s opinions are outside $\epsilon$ distance of each other.

**$k$-edge-connectivity/min-max flow.** The simplest method to calculate the $k$-edge-connection of a graph is to turn the problem into a series of maximum flow problems. We find the minimum of all the maximum flow problems, hence the min-max flow algorithm. The maximum flow problem is defined as follows. Let $G = [V, E]$ be a weighted digraph. Let there be a

source vertex $s \in V$ and a sink vertex $t \in V$. The weight of each edge is its capacity $c \in \mathbb{R}$. A flow is a function $f : E \rightarrow \mathbb{R}$ which satisfies these conditions.

- Capacity constraint: The flow over an edge must not exceed its capacity $c$.

- Conservation of flows: The flow entering a vertex must equal the flow leaving the vertex, excluding $s$ and $t$.

The maximum flow problem is to route as much flow from $s$ to $t$, which gives the maximum flow rate $f_{max}$. Algorithms to find $f_{max}$ are the Ford–Fulkerson algorithm [27], Dinic's algorithm [28] and push–relabel algorithm [29]. See [29] for a more extensive list of algorithms. In this paper we use the MatLab built-in `flow` function to find $f_{max}$.

To apply a maximum flow algorithm to an unweighted and undirected graph $G$, we need to convert $G$ to a weighted digraph. We accomplish this by replacing every edge in $G$ with two directed edges. The two directed edges connect the two previously connected vertices. Lastly, we assign the capacity of the new directed edges to be one. We find the $k$-edge-connectivity of $G$ using maximum flow by first iterating over every pair of vertices. We set one vertex as the source and the other as the sink and find $f_{max}$ for that source and sink pair. The minimum of those $f_{max}$ will be the $k$-edge-connectivity of $G$.

The algorithm to find $k$-edge-connectivity follows from Menger's theorem, which is a special case of the max-flow min-cut theorem [27], stating that the number of edge independent paths between two vertices is equal to the minimum set of edge cuts that separate those two vertices. Therefore finding the minimum of the $f_{max}$ derived from the directed version of $G$ will result in $k$-edge-connectivity of $G$ [30]. Noted in [31] we can improve the algorithm by fixing a vertex and finding the minimum of its maximum flows with all other vertices in the graph.

The adjacency matrices created by the Martins model [2, 16] are weighted digraphs. Applying the min-max flow algorithm will result in a meaningful connectivity measurement, hence polarisation measurement, even with a non-integer result for '$k$'. H-K bounded confidence model produces an undirected and unweighted graph and gives $k \in \mathbb{Z}$. So there will be no difficulties in applying the min-max flow. Although more efficient algorithms exist for finding $k$-edge-connectivity of an unweighted graph, for consistency, we will still use the min-max flow algorithm since these algorithms won't work on the Martins' adjacency matrix. See S3 Appendix in S1 File for other potential methods to calculate min-max flow.

**Largest cluster size with spectral radius.** Determining the spectral radius was simple. After producing the adjacency matrix at a particular time in the simulation, we calculated the spectral radius of the adjacency matrix using the in-built Matlab function `eig`.

## Applying the *f*-divergences measures

The general approach with the *f*-divergences was to determine the pairwise divergences between all agents and then measure the mean *f*-divergence.

**Kullback–Leibler divergence.** Applying Kullback–Leibler divergence to the Martins model is simple. Since the Martins model considers opinions as normal distortions, finding the Kullback–Leibler divergence between two agents is finding the K-L divergence between two Gaussians *f* and *g*. This simplifies Eq 2 to

$$\text{KLD}(f||g) = \log\left(\frac{\sigma_j}{\sigma_i}\right) + \frac{\sigma_i^2 + (x_i - x_j)^2}{2\sigma_j^2} - \frac{1}{2}, \tag{6}$$

where $x_i$ and $\sigma_i$ are the mean and standard deviation for *f*, and $x_j$ and $\sigma_j$ are the mean and

standard deviation for $g$ (for deviation see S4 Appendix in S1 File). Using Eq 6, we can calculate the KLD between all possible agent pairs and then calculate the mean KLD. Agents in an H-K bounded confidence model simulation have very definitive opinions (i.e. places where they rank other opinions as 0), creating singularities in K–L divergence; thus, we can't use K–L divergence to measure the polarisation of those simulations. K–L divergence is thereby limited in its applicability which we discuss in the discussion section of this paper.

**Hellinger distance.** It is simple to apply Hellinger distance to the Martins model. Since every agent's opinion is essentially a normal distribution, we can find the squared Hellinger distance between two normal distributions, which is

$$H^2 = 1 - \sqrt{\frac{2\sigma_1\sigma_2}{\sigma_1^2 + \sigma_2^2}} e^{\frac{(x_1 - x_2)^2}{4(\sigma_1^2 + \sigma_2^2)}},$$

where $x_1$ and $x_2$ are the means and, $\sigma_1$ and $\sigma_2$ are the standard deviations of two normal distributions [32]. Applying the Hellinger distance to the H-K bounded confidence model is less trivial. Although agents don't have probability density functions for opinions, we can consider an agent's opinion as a uniform distribution over $[x_i - \epsilon, x_i + \epsilon]$. The Hellinger affinity of two agents' opinions will be the area of overlap between the two uniform distributions. The squared Hellinger distance is

$$H^2(f, g) = \begin{cases} \dfrac{x_1 - x_2}{2\epsilon} & \text{if } x_1 - x_2 \leq 2\epsilon \\[2mm] 1 & \text{if } x_1 - x_2 > 2\epsilon \end{cases}$$

where without loss of generality we assume that $x_1 > x_2$ (for deviation see S5 Appendix in S1 File).

Like with KLD, we can find the Hellinger distance between all agents and then find the mean of those Hellinger distances to use as a measure of polarisation. Finding the mean of these $f$-divergences provides a general perspective on the differences between agents, thus providing resistance to outlying agents.

## Results

This section presents the results of simulations from Martins and the H-K bounded confidence models for different initial conditions and model parameters and analysed polarisation using the measures based on the min-max flow rate, spectral radius, and mean KLD and Hellinger distance. We varied the parameter $\epsilon$ for the H-K bounded confidence model. We varied the initial $\sigma$ and fixed $p$ at 0.7 for the extended Martin's model. For each set of initial conditions, we ran 100 simulations, each consisting of $n = 1000$ agents. Figs 1 and 2 shows a sample of simulation output for each initial condition.

### Cluster counting and $y$

We calculated the cluster counts and $y$-statistic for every simulation to compare with the new methods we developed in this paper. Tables 1 and 2 show the mean cluster count for the simulations. Both tables reveal the relationship we expect to see between the cluster count and the parameter values. The cluster count for a simulation is inversely proportional to both the initial uncertainty for the Martins model and $\epsilon$ for the HK bounded confidence model. Fig 3 shows the $y$-statistic for the simulations. We note that the $y$-statistic seems to categorise simulations as either polarised or in consensus. Interestingly in Fig 3B, the $y$-statistic can identify that simulations are closer to consensus for initial uncertainty of 0.14.

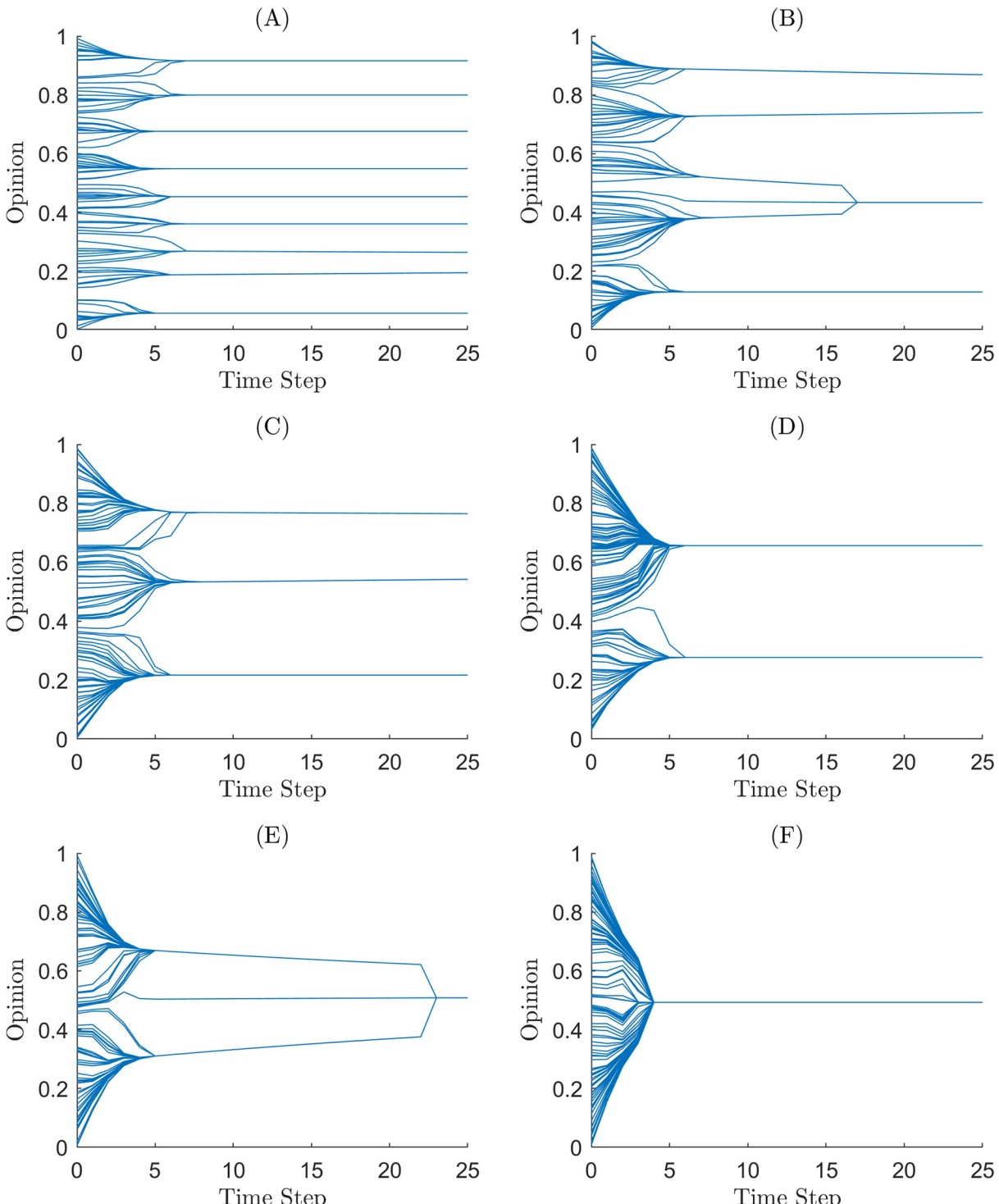

**Fig 1. Sample simulations' opinion shifts of agents through time for the HK bounded confidence model under different values of $\epsilon$.** (A) A simulation with $\epsilon = 0.05$. (B) A simulation with $\epsilon = 0.1$. (C) A simulation with $\epsilon = 0.15$. (D) A simulation with $\epsilon = 0.2$. (E) A simulation with $\epsilon = 0.25$. (F) A simulation with $\epsilon = 0.3$.

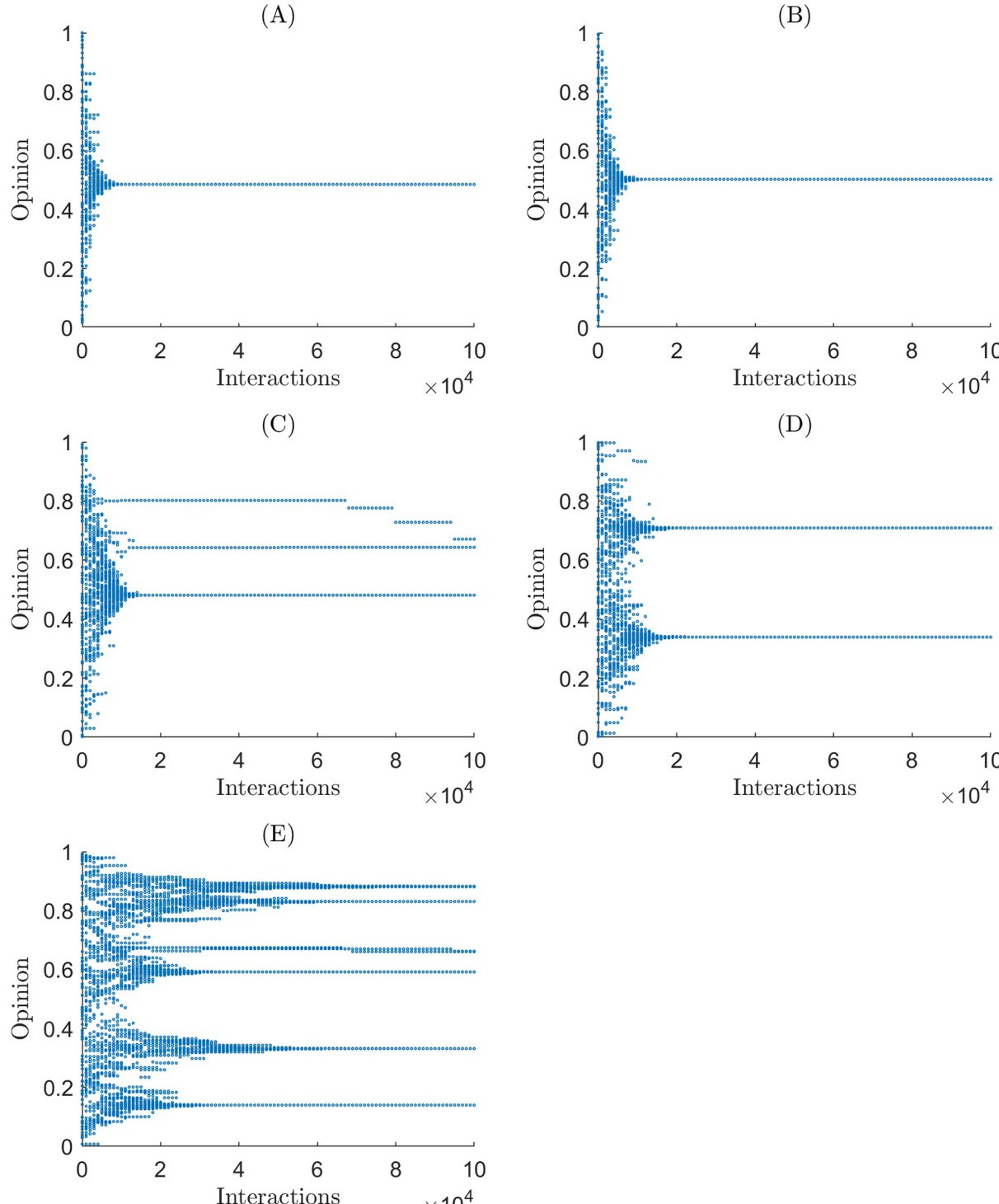

**Fig 2. Sample simulations' opinion shifts of agents through time for the extended Martins model under different initial uncertainty.** (A) A simulation with an initial $\sigma = 0.5$. (B) A simulation with an initial $\sigma = 0.2$. (C) A simulation with an initial $\sigma = 0.14$. (D) A simulation with an initial $\sigma = 0.1$. (E) A simulation with an initial $\sigma = 0.05$.

**Table 1. The mean cluster count for each 100 simulations of the HK bounded confidence under different values of $\epsilon$.**

| $\epsilon$ | 0.05 | 0.1 | 0.15 | 0.2 | 0.25 | 0.3 |
|---|---|---|---|---|---|---|
| Mean Cluster Count | 7.52 | 3.74 | 2.64 | 1.99 | 1 | 1 |
| Standard Deviation | 0.6432 | 0.4845 | 0.4824 | 0.1 | 0 | 0 |

Cluster counts were found using the inbuilt Matlab function `subclust`.

**Table 2. The mean estimated opinion cluster count for each 100 simulations of the extended Martins model under different initial uncertainty.**

| Initial Uncertainty | 0.5 | 0.2 | 0.14 | 0.1 | 0.05 |
|---|---|---|---|---|---|
| Mean Cluster Count | 1 | 1 | 1.15 | 2.03 | 4.18 |
| Standard Deviation | 0 | 0 | 0.3589 | 0.1714 | 0.73 |

Cluster counts were found using the inbuilt Matlab function `subclust`.

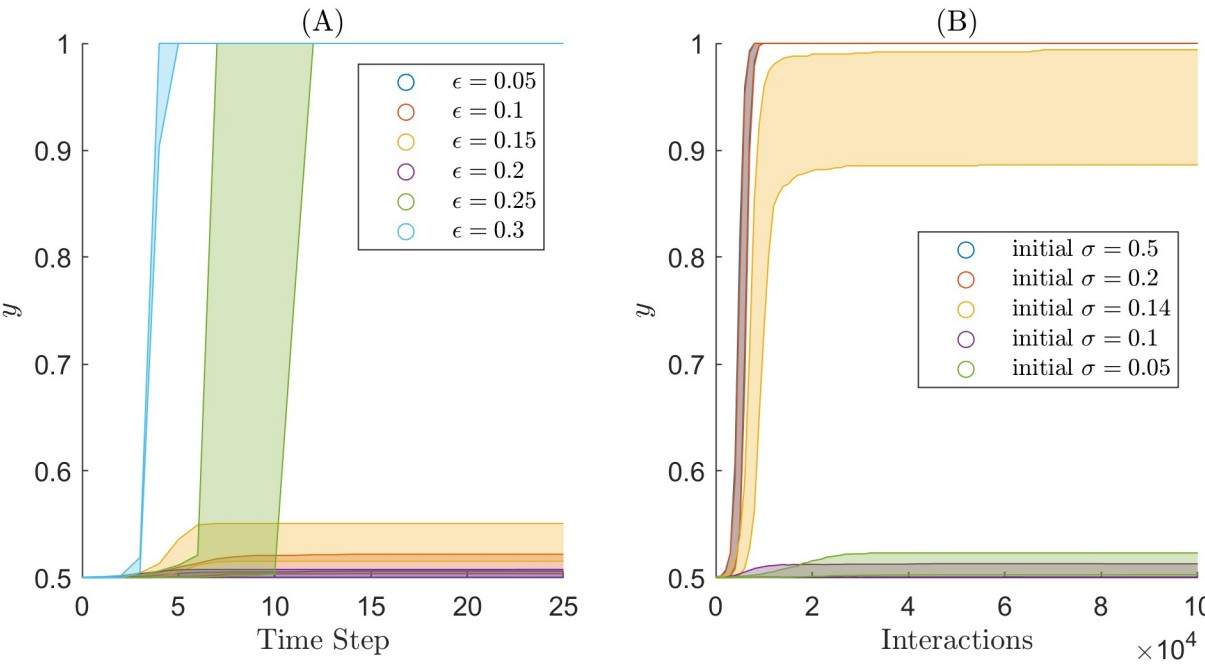

**Fig 3. The *y*-statistic of simulation for the two different models and different parameter values.** (A) HK bounded confidence simulations. (B) Extended Martins model simulations.

## The min-max flow/edge connectivity

The min-max flow rate is a bounded measure of polarisation which diverges to 0 at polarisation or $n-1$ at a consensus, where $n$ is the number of agents in the simulation. Finding the min-max flow rate or edge connectivity is the most computationally intense method of measuring polarisation, and measures of min-max flow rate over individual simulations are noisy, regardless of simulation size $n$. Thus averaging over the 100 simulations produced constant behaviour as an illustration of the method's utility.

Fig 4 shows the edge connectivity of the H-K bounded confidence model at different values of $\epsilon$. For $\epsilon \leq 0.2$, the edge connectivity diverges to 0 at the steady-state, indicating that the

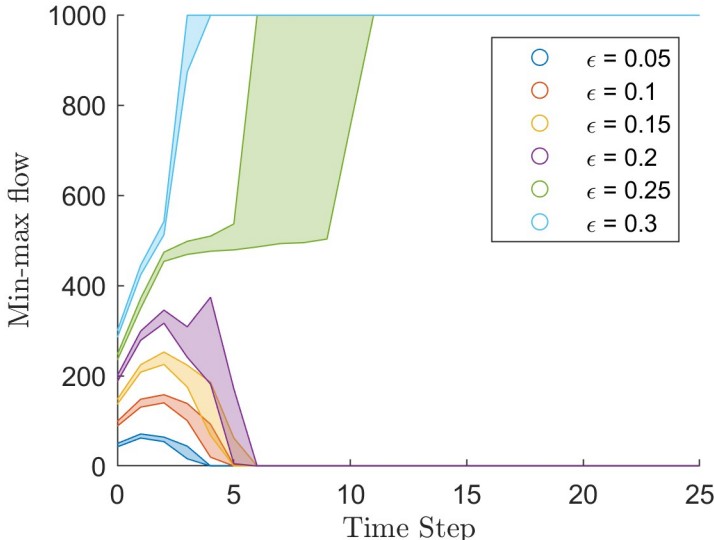

**Fig 4. Min-max flow, _k_, through time of the simulations that used the H-K bounded confidence model with different values of $\epsilon$.**

resulting graph is disjoint, and the simulation has polarised. Note some outlying simulations where $\epsilon = 0.2$ converged to consensus.

For values of $\epsilon > 0.2$, the edge connectivity increased to _n_, suggesting consensus (i.e. every node directly connects to every other node). The $\epsilon = 0.25$ took longer to reach consensus (five to ten iterations) compared to the three iterations when $\epsilon = 0.3$. The longer time for the $\epsilon = 0.25$ simulation to reach consensus suggests that when $\epsilon \approx 0.25$ simulations can transition between consensus and polarisation.

Fig 5 shows the edge connectivity of the Martins confidence model at different values of initial $\sigma$. For initial $\sigma \leq 0.14$ edge connectivity decreased to 0, which contrasts with Fig 4 where

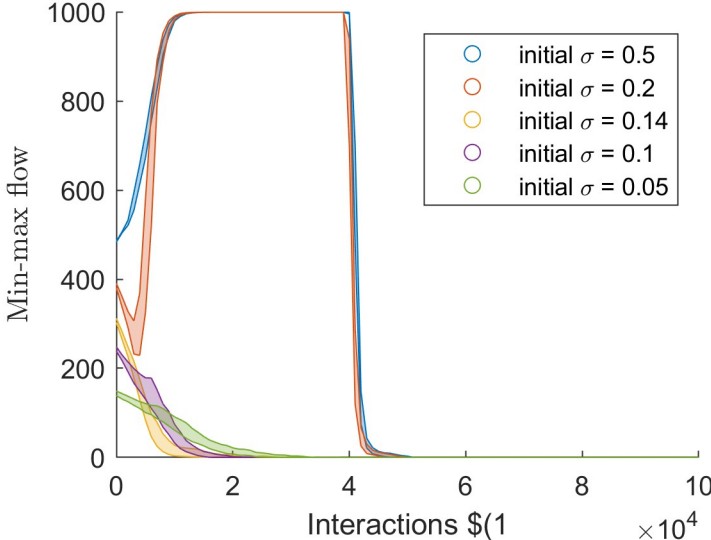

**Fig 5. Min-max flow, _k_, through time of the simulations that used the Martins model with different values of initial uncertainty.**

edge connectivity reached a local minimum. To further compound this difference, Fig 4 shows that the larger values of $\epsilon$ bound the edge connectivity of smaller values. Such a pattern does not exist in Fig 5.

For initial $\sigma > 0.2$ edge connectivity increases to 1000, but at initial $\sigma = 0.2$ the progression to 1000 is not monotonic. Eventually, the edge connectivity of all simulations fell to 0. This behaviour is consistent with the analysis from [2], where the model, in the long term, was demonstrated to approach consensus arbitrarily close before fragmenting into opinion clusters. Edge connectivity decreases to a local minimum for the first few thousand interactions before increasing to 1000. Of note is that in the lower 25% quartile of simulations, with initial $\sigma = 0.2$, edge connectivity continued decreasing and reached 0. Falling to a local minimum when $\sigma = 0.2$ suggests that $\sigma = 0.2$ is close to a bifurcation point between polarisation and consensus.

## Spectral radius

Fig 6 displays the behaviour of the spectral radius in the H-K bounded confidence model. In most cases, simulations reach a steady-state value of spectral radius after three iterations of the simulation, the exception being $\epsilon = 0.25$ and $\epsilon = 0.15$. At $\epsilon = 0.25$, the simulations first converge to a spectral radius of 500, but for some simulations, after 2–7 iterations, the spectral radius jumps to 1000. At $\epsilon = 0.15$ the simulations' spectral radius converges at either where $\epsilon = 0.1$ converges or $\epsilon = 0.2$ converges.

Fig 7 displays the spectral radius of the extended Martins model under several different parameterisations. The results show two phases of behaviour for the simulations' spectral radii. First, the simulations reach steady-state, and second, they begin to fragment in their opinion clusters. There is little variability for initial $\sigma$ of 0.5 or 0.2, where the simulations reached consensus, except when the simulations enter the second phase, where the spectral radii vary greatly. At initial $\sigma$ 0.1 and 0.05, the simulations polarised and took longer to reach the second phase. In the second phase, all simulations seem to drop between the same value in

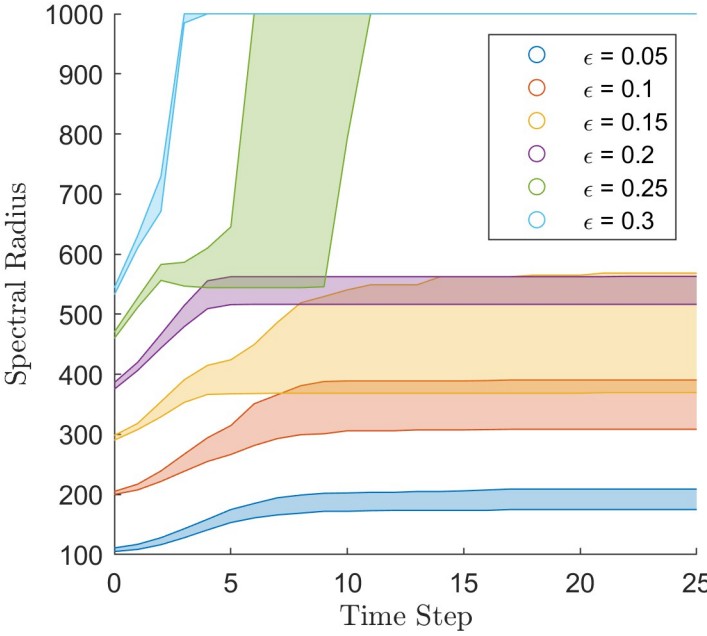

**Fig 6. Spectral radius through time of the simulations that used the H-K bounded confidence model with different values of $\epsilon$.**

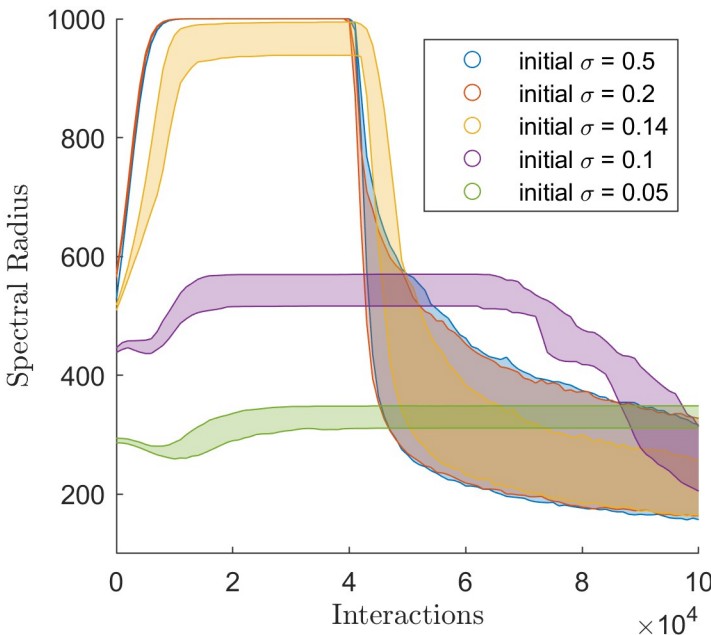

**Fig 7. Spectral radius through time of the simulations that used the Martins model with different values of initial uncertainty.**

spectral radius, between 150 and 350. Potentially there is some structure to how agents fragment in the extended Martins model.

What is of particular interest is the results for initial $\sigma = 0.14$. In [2], initial $\sigma = 0.14$ was the critical value at which the simulations polarised, when one opinion cluster turned into two. The results in Fig 7 show that at initial $\sigma = 0.14$, rather than dividing into two even opinion clusters (as implied in [2]), a small portion of agents (approximately 10–100) break away to form their cluster while the rest remain in consensus. The spectral radius highlights the continuous transition between consensus and polarisation while counting opinion clusters obfuscates this behaviour, although the $y$-statistic and averaging over 100 simulations can identify this behaviour.

### Mean K-L divergence

Fig 8 shows the behaviour of the mean K-L divergence in the Extended Martins Simulations. During the beginning of the simulations, K-L divergence inversely correlates with the initial $\sigma$. Later in the simulation, K-L divergence grows exponentially. There are two phases to the mean K-L divergence in Fig 8, similar to the spectral radius. In the first phase, K-L divergence grows at a fixed exponential rate which we observe as a linear trend in Fig 8. The next phase has that fixed exponential growth rate decrease. These phases are present in all of the simulations. The transition between the first and second phases appears to happen at the same amount of divergence. For $\sigma \geq 0.2$, there is no exponential growth until the simulations start fragmenting, and then the simulation seems to pick a random rate of exponential expansion.

**Interpreting the exponential expansion.** There is a link between the rate of exponential expansion and the number of opinion clusters in a simulation since the exponential expansion rate correlates with initial $\sigma$ (except for simulations which reached consensus), which then is inversely correlated with the number of opinion clusters as seen in Fig 1. We shall now develop this further in this section.

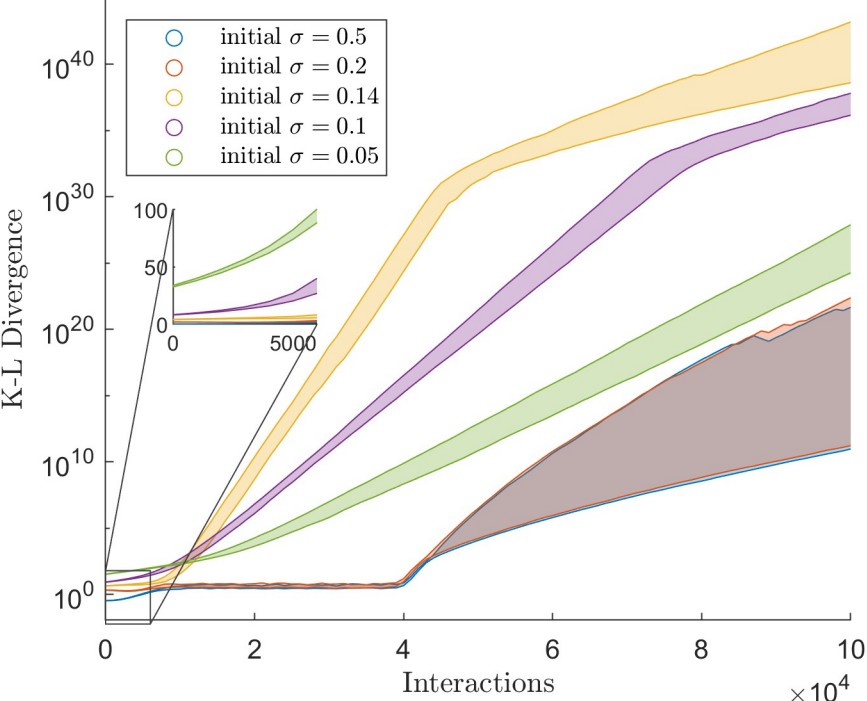

**Fig 8. Mean K-L divergence through time of the simulations that used the Martins model with different values of initial uncertainty.**

The reason for the exponential growth of $\overline{\text{KLD}}$ is because this term

$$\frac{\sigma_i^2 + (x_i - x_j)^2}{2\sigma_j^2} \tag{7}$$

in Eq 6. In the late stages of the Martins model, when agents successfully interact, both agents will half their 'variance' (uncertainty squared), causing the KLD they share with other agents to double. We can use this fact to estimate the number of clusters in a simulation by fitting a linear regression to the $\log(\overline{\text{KLD}})$. Where $m$ is the gradient of the linear regression and $n$ is the number of agents in a simulation, we found the general expression for the effective estimated cluster count to be

$$\hat{\psi} = \frac{\log(n+2) - \log(n)}{m}. \tag{8}$$

S6 Appendix in S1 File provides a more detailed deviation of Eq 8.

Table 3 shows the result of applying Eq 8 for simulations that generated more than one opinion cluster. These estimates are close to the number of opinion clusters in Table 2 for their appropriate initial uncertainties. Moreover, the variance is significantly lower than in Table 2.

In practice, Fig 8A shows that simulations which reach consensus have no exponential growth, hence $m = 0$, and the derivation of Eq 8 no longer applies.

**Table 3. The mean estimated opinion cluster count for each 100 simulations under different initial uncertainty.**

| Initial Uncertainty | 0.14 | 0.1 | 0.05 |
|---|---|---|---|
| Mean $\hat{\psi}$ | 1.1055 | 1.91751 | 3.8231 |
| Standard Deviation | 0.1644 | 0.1773 | 0.5745 |

## Mean Hellinger distance

The mean Hellinger distance behaves similar to the spectral radius. The mean Hellinger distance results mirror the same features found in the spectral radius results. The mean Hellinger distance in general varied less across simulations.

Fig 9 shows the mean Hellinger distance across the various H-K bounded confidence simulations. The mean Hellinger distance converges within five time steps of a simulation with expectations for $\epsilon = 0.25$, converging within ten time steps. As discussed in the spectral radius results section, $\epsilon = 0.25$ is close to a tipping point between polarisation and consensus. The mean Hellinger distance varies more when there is less polarisation, excluding when simulations reach consensus. At $\epsilon = 0.05$, there is little variation in mean Hellinger distance between simulations, whereas, at $\epsilon = 0.20$, there is more variation in mean Hellinger distance between simulations. We postulate that when a simulation splinters into many opinion clusters, the average distance between clusters remains constent, whereas, when there are only two clusters, those two clusters can be close or on opposite ends of opinion space.

Fig 10 shows the mean Hellinger distance across the various Martins simulation simulations. The mean Hellinger distance for Martins largely follows the behaviour of spectral radius for Martins. The mean Hellinger distance differs in one way from the spectral radius. No simulations reached a 'consensus' with mean Hellinger distance measuring polarisation. The closest a simulation comes to consensus is a mean Hellinger distance of above 0.1. Initially,

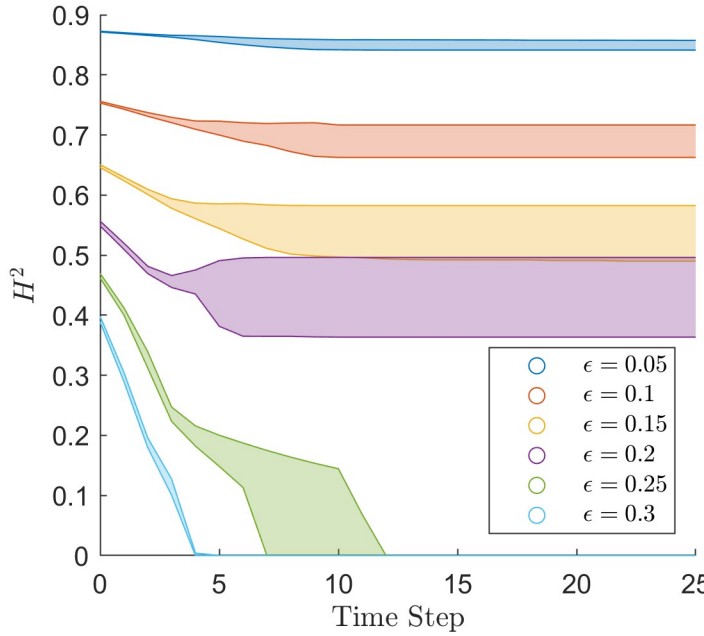

**Fig 9. Mean Hellinger distance through time of the simulations that used the H-K bounded confidence model with different values of $\epsilon$.**

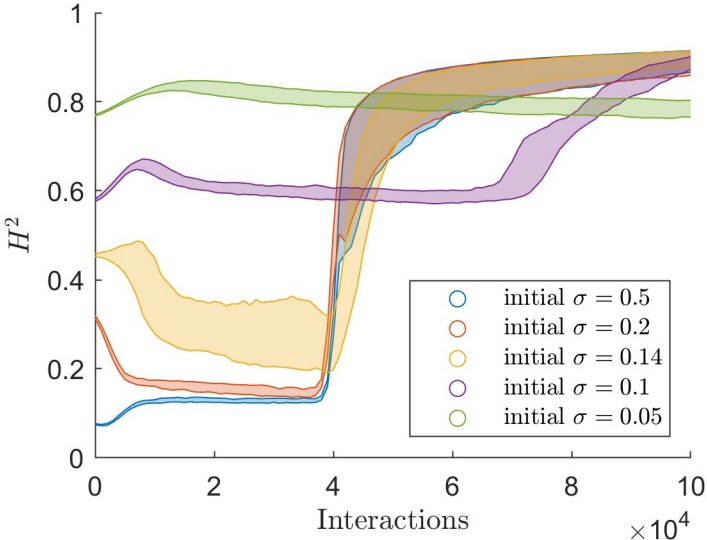

**Fig 10. Mean Hellinger through time of the simulations that used the Martins model with different values of initial uncertainty.**

simulations with initial $\sigma = 0.5$ began with a mean Hellinger distance below 0.1, which converged to a value above 0.1.

## Discussion

This paper investigated four methods of determining polarisation, the min-max flow rate, spectral radius, the mean Kullbeck-Liebler divergence and the Hellinger distance. The spectral radius and min-max flow methods use graph theory concepts to form the basis for measuring polarisation. As a consequence, both have physical interpretations based on network topology. The min-max flow rate is the minimum number of paths between all vertex pairings. The spectral radius relates to the largest size of an opinion cluster, among other graph theory concepts such as the number paths length $k$. As a result, both measure different aspects of the connectivity of a graph, and through their different approaches, we can uncover more understanding of polarisation.

The min-max flow rate is limited as a dichotomous measurement of polarisation and is useful when the network is not in a steady state for tracking the trajectory of polarisation. The min-max flow rate is a non-robust measure, as it is sensitive to outliers. We would need to prune real-world networks to remove outliers to use the min-max flow rate as a measure of polarisation. The advantage of the min-max flow rate is that its definition is conceptually straightforward to interpret. Spectral radius is more versatile than min-max flow rate and is more robust to outliers, making it more practical to apply to real-world networks. As a non-dichotomous measure, the spectral radius can be used to monitor trajectories of polarisation and as a comparison between networks in a stead-state. The drawback of using spectral radius as a measure of polarisation is that its derivation is more challenging and esoteric.

The Kullbeck-Liebler divergence (KLD) is an information-theoretic measure of information loss (or gain). The KLD measures the divergence between two probability density (or mass) functions. In the Martins model, individual opinions are represented as probability density functions, leading to the KLD as a natural measure to consider as the difference or distance between opinions. The mean KLD is the average of all pairwise KLD for a network. It is

important to note that, unlike the min-max flow rate and spectral radius, the mean KLD doesn't account for network topology (min-max flow rate and spectral radius make use of information about the individuals' opinions via the emergent social network structure). Therefore, the mean KLD is limited to circumstances where opinions are probability distributions. What limits the mean KLD further is that it can't handle definite probability distributions. It is impossible to meaningfully calculate the mean KLD for the HK bounded confidence model because it requires the calculation of the KLD between two uniform distributions resulting in an infinite KLD.

We noted that the exponential growth rate of the KLD (Eq 13 in S6 Appendix of S1 File) after the simulation achieved steady-state is inversely proportional to the number of clusters at steady-state. Thus the slope of the log-linear model of a simulation, $m$, can be used as a measure of polarisation by producing an effective cluster count. The effective number of clusters $\hat{\psi}$ from (8) agrees with the spectral radius interpreted polarisation, including the measured cluster counts in Table 2. For example, given a simulation with $\sigma = 0.14$, $\hat{\psi} = 1.1055$ indicating that the agents are "mostly" in consensus, but that there is still some disagreement. The conclusions of this measure match the conclusions drawn from the spectral radius and what we observe in Fig 2 and Table 2, where simulations converge to ("mostly") a single cluster. Compared with the spectral radius, which describes the size of the largest cluster, $\hat{\psi}$ is more comprehensive as it describes the effective number of clusters for the simulation at steady-state. A weakness of $\hat{\psi}$ is that it is ill-defined when the actual cluster counts $\psi = 1$. As seen in Fig 8 the two types of simulations that reached consensus exhibited no exceptional growth when they first reached steady-state, making $m = 0$, which breaks Eq 8 making $\hat{\psi} \to \infty$. Considering this instability only occurs when a simulation reaches consensus, it is easy to ignore since we can identify consensus visually. Although, simulation types close to consensus will inherit some instability since some of the simulations will fall into consensus through random chance. Of more pressing concern is that $\hat{\psi}$ tied in with the Martins model and KLD, which could limit the applicability of $\hat{\psi}$ to more realistic situations if the Martins model does not reflect how individuals share opinion. Still $\hat{\psi}$ hints at the possibility of a continuous extension to counting clusters.

The mean Hellinger distance is similar to the spectral radius. The only significant deviation from the spectral radius is that the mean Hellinger distance never reached zero in the Martins simulations. Because the Hellinger distance is relative to uncertainty (Eq 3), like the Martins updating rules, it could determine that Martins simulations were never in complete agreement. This novel ability suggests that the Hellinger distance has an advantage over the spectral radius. Where the spectral radius might determine a group to be in complete consensus, the Hellinger distance can correctly determine that the group is not in complete consensus.

The spectral radius of the H-K Bounded confidence in Fig 6 is consistent with the results in [3], showing that in the homogeneous case, the model forms uniformly spaced opinion clusters, with the space between them being greater than $\epsilon$. For the H-K bounded confidence model, the spectral radius reflects this discrete nature of opinion clusters, Fig 6 shows the spectral radius converging to quantised values depending on $\epsilon$. The spectral radius can track simulations when they fall between two quantised states. We can see in Fig 1 that simulations with values of $\epsilon = 0.15$ and $\epsilon = 0.25$ have opinion cluster merge later in the simulation. This is reflected in the spectral radius through Fig 6 as the spectral radius at $\epsilon = 0.15$ and $\epsilon = 0.25$ has more variability and stretch over the neighbouring values of $\epsilon$. So the spectral radius can determine bifurcation points in parameter values, i.e. parameter values at which two opinion clusters merge into one.

Of particular note is the late stage behaviour of Figs 7, 8 and 10. The Martins model is known to fragment at the late stages of a simulation, but the fragmentation doesn't result in complete disunity (where every agent is isolated from all other agents). From Fig 7 the simulations drop to a spectral radius between 200 and 400. It would be interesting to investigate the social network in the late stages of the Martins model.

## Conclusion

This paper has investigated four methods of measuring polarisation. We conclude that the min-max flow rate is the most insufficient method. The main advantage of the method is that it is intuitive. Although the min-max flow rate does reveal some dynamics as a simulation falls into either polarisation or consensus, simulations can only be in consensus or polarisation. Overall the method at most performs equivalently to the $y$-statisitc. The method is also extremely sensitive to the outlying agent, which is a problem in any real-world application. The major complication with the method is the computation time which makes the method less useful. Although, we briefly investigated a method to improve computational efficiency that resulted in a new measure of polarisation (see S3 Appendix in S1 File).

In contrast, the spectral radius provided a complete picture of polarisation. The method's physical meaning is loose and difficult to understand, but we understand its meaning as the effective largest cluster size in a simulation, i.e. what the largest cluster would be if the simulation reached steady-state. As a result, the spectral radius places polarisation on a continuum and can identify when a simulation is close to reaching a consensus. Furthermore, the method can distinguish between different levels of polarisation, i.e. three cluster simulations from two cluster simulations. Essentially the spectral radius blends cluster counting methods and the $y$-statistic together. So we consider the spectral radius an effective at measuring polarisation.

The mean K-L divergence is interesting because the results were initially difficult to interpret. The mean K-L divergence diverged to infinity exponentially, but from the exponential expansion rate, we could determine the 'number of clusters' $\psi$ of a particular simulation. The estimated cluster number agreed with what we observed with other measures and the raw results. It is clear that $\psi$ is what we can use to measure polarisation, and it is more intuitive to grasp compared to the spectral radius, but the measure has its drawbacks. First, it breaks when a simulation reaches consensus and second, it relies on the K-L divergence and the mechanics of the Martins model, which in real-world applications might not hold. Still, this method promises a way to express opinion cluster number as a continuous value which would be another avenue of research.

The mean Hellinger distance closely resembles the spectral radius but differs from the spectral radius in one crucial way. The Hellinger distance is more sensitive at complete consensus; it had more foresight into the Martins model degeneration from consensus into arbitrarily close opinion clusters than other methods. The mean Hellinger distance has an advantage over the spectral radius with its ability to detect the finer dynamics of the Martins model sooner.

Overall these four methods measure different aspects of polarisation, and the individual measures fail to capture the whole process of polarisation, but together they reveal the complete picture. Depending on the circumstances, certain methods might be more effective than others. The spectral radius is the most general and can be applied in most situations, whereas the mean Hellinger distance and K-L divergence work better when applied to their appropriate niches.

In this paper, we have only looked at simulated societies. Future research would involve applying these measures to real data sets. It is clear from the investigations in this paper that the mean K-L divergence, Hellinger distance and spectral radius hold the most promise.

## Supporting information

**S1 Fig. Alternate min-max flow compared with min-max flow for HK bounded confidence.**
(TIF)

**S1 File.**
(PDF)

## Author Contributions

**Conceptualization:** Johnathan A. Adams, Gentry White.

**Data curation:** Johnathan A. Adams.

**Formal analysis:** Johnathan A. Adams, Gentry White.

**Funding acquisition:** Johnathan A. Adams.

**Investigation:** Johnathan A. Adams.

**Methodology:** Johnathan A. Adams.

**Project administration:** Gentry White, Robyn P. Araujo.

**Software:** Johnathan A. Adams.

**Supervision:** Gentry White, Robyn P. Araujo.

**Visualization:** Johnathan A. Adams.

**Writing – original draft:** Johnathan A. Adams.

**Writing – review & editing:** Johnathan A. Adams, Gentry White, Robyn P. Araujo.

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
