## [Decision Letter · Decision Letter 0]

7 Jun 2022

PONE-D-22-04726

Mathematical measures of societal polarisation

PLOS ONE

Dear Dr. Adams,

Thank you for submitting your manuscript to PLOS ONE. After careful consideration, we feel that it has merit but does not fully meet PLOS ONE’s publication criteria as it currently stands. Therefore, we invite you to submit a revised version of the manuscript that addresses the points raised during the review process.

Please carefully revised your manuscript according to the reviewers suggestions.

We look forward to receiving your revised manuscript.

Kind regards,

Ismael González Yero

Academic Editor

PLOS ONE

**Journal requirements:**

Reviewers' comments:

Reviewer's Responses to Questions

**Comments to the Author**

1. Is the manuscript technically sound, and do the data support the conclusions?

Reviewer #1: Yes

Reviewer #2: Yes

2. Has the statistical analysis been performed appropriately and rigorously? 

Reviewer #1: N/A

Reviewer #2: Yes

3. Have the authors made all data underlying the findings in their manuscript fully available?

Reviewer #1: Yes

Reviewer #2: No

4. Is the manuscript presented in an intelligible fashion and written in standard English?

Reviewer #1: Yes

Reviewer #2: No

5. Review Comments to the Author

Reviewer #1: This manuscript proposes four measures of polarization and the measures studied using two models of opinion formation. I only spotted minimial technical mistakes that coule easily be corrected, so I therefore recommend it for publication in PLOS ONE, with minor revision.

Technical comments:

* "Sub-graph" is imprecise on p.3. From the context, it seems like what was meant is "connected components." A subgraph is any graph induced by taking a subset of nodes. In the same vein, "Weakly connected" is precisely defined in graph theory and has to do with reachability in directed graphs. I would use a different term, say "sparsely connected."

* In the literature review: It is a bit weird to disparage counting methods when, in the end, spectral analysis is mainly sold as a counting method.

* Spectral analysis: The radius will only be n if the diagonal is set to 1, but usually, one would set the diagonal to zero for an adjacency matrix (a node is not connected to itself).

* Eq (4): There is a typo. One should read (1-p^*) right after the equality.

* P8: In theory, one could also think of the HK opinions as "uniformly distributed" when applying the KLD. So why only do it for Hellinger?

* Fig.1: The resolution of the figure was quite low on my computer. Consider using a better resolution in the final version.

* Exponential expansion: Isn't the exponential expansion damning of the KLD measure? Visually the opinions have clustered (Fig. 1), and only the lack of variance leads to many opinions. This leads to several counter-intuitive results (see also the second "phase" of Fig3 and 5).

* p14: I don't think that interpreting the spectral radius as the size of the component is correct. It also captures connectivity to some extent. For example, the radius will slowly transition from ~n/2 to n-1 if we start with two disconnected, fully connected blocks and slowly connect them until we have a single block where all connections exist.

* The way the network structure is defined is a missed opportunity, in my opinion. The graph structure is wholly derived from the opinion values themselves. Hence it does not add a second layer of data (e.g., a social network on which opinions are exchanged). In the end, the proposed graph measures could largely be computed without references to the graph (e.g., for the min-cut and the HK model: slide a line across the opinion line and find the location such that there is the least agents within a fixed band. Report the number of agents as the size of the cut.)

Reviewer #2: The research paper entitled “Mathematical measures of societal polarisation: Graph and information theory Approaches,” is very well-framed and well-organized along with adequate details and descriptions. The results are promising and easy to follow. However, there are few comments that needs to be addressed before it can be recommended for publication in the journal of the Plos One.

1. The main innovation of this manuscript is limited and unclear. The reviewer suggests that the authors spend a great amount of effort to better clarify the novelty of their proposed technique. In overall, this manuscript lacks accurate discussions on the differentials of the proposed method. Why is this proposed technique better than others? What exactly are the contributions of the proposed technique? The reviewer suggests the following approaches to solve this issue: comparisons could be made by implementing and running other techniques over the same raw data. Such a comparison can be made in terms of efficacy or efficiency (how good the proposed technique is in using the available resources - i.e. time, computation...).

2. Some figures, especially figure #1, shown in this manuscript look blurred; therefore, the authors are suggested to improve their quality and legibility.

3. In the abstract, it would be better to highlight and specifically stress what has been done in this research work. The present abstract fails to accurately reflect the novelty of the proposed framework.

4. In the introduction section, many recent and relevant contributions close to the authors’ research work are not reviewed. The authors should consider and mention some more recent and relevant references to the literature review to create a proper perspective of the existing state-of-the-art in the area of interest.

5. More details and references concerning Kullback–Leibler divergence and the Hellinger distance metrics are required. Moreover, the paper shows a lack of references. Some equations and formulations in the text need references and/or mathematical demonstrations.

6. There are some English typos throughout the manuscript (especially, in the abstract, introduction, and conclusion). The authors are needed to thoroughly revise the paper and correct the English grammar mistakes.

6. PLOS authors have the option to publish the peer review history of their article (what does this mean?). If published, this will include your full peer review and any attached files.

Reviewer #1: No

Reviewer #2: No

---

## [Author Response · Author response to Decision Letter 0]

22 Jul 2022

Reviewer #1:

* "Sub-graph" is imprecise on p.3. From the context, it seems like what was meant is "connected components." A subgraph is any graph induced by taking a subset of nodes. In the same vein, "Weakly connected" is precisely defined in graph theory and has to do with reachability in directed graphs. I would use a different term, say "sparsely connected."

Response: The reviewer is correct in assuming that we meant the term “connected components.” We also agree that “sparsely connected” is more accurate to our usage than “Weakly connected.” We have updated the manuscript to have the correct terminology.

* In the literature review: It is a bit weird to disparage counting methods when, in the end, spectral analysis is mainly sold as a counting method.

Response: The advantage of the spectral radius is that it provides a more granular counting method. The spectral radius represents the size of the largest cluster of a given simulation, and if the largest cluster size is close to the number of agents, we can conclude that the simulation is close to consensus. Whereas with counting methods, the closest a simulation can be to polarisation is when it has two clusters, which leaves little room to distinguish between simulations that are close to consensus or bi-polarised and, by extensions, makes it hard to determine how polarised societies are.

* Spectral analysis: The radius will only be n if the diagonal is set to 1, but usually, one would set the diagonal to zero for an adjacency matrix (a node is not connected to itself).

Response: True, but the adjacency matrices we are investigating describe interaction potential, i.e. how much an agent will accept another agent if they interact. The self-interaction potential measures how much an agent accepts another agent's opinions if the other agent's opinion is identical to their own opinion. In the case of the Martins Model, it would measure how stubborn an agent is, contributing to the overall strength of an opinion cluster. In the case of the HK model, an agent's initial opinion contributes to their new opinion equal in importance to any other agents within ε of the agent. Moreover, the French model requires agents to interact through adjacency matrices which describe graphs with self-loops, and the weighting of the self-loop is a measure of an agent's stubbornness. We have made exploratory computations into adjacency matrices that omit self-loops and found that the spectral radius is one less than when including self-loops.

* Eq (4): There is a typo. One should read (1-p^*) right after the equality.

Response: We have fixed the typo the reviewer has identified.

* P8: In theory, one could also think of the HK opinions as "uniformly distributed" when applying the KLD. So why only do it for Hellinger?

Response: We initially tried the same inquiry the reviewer described, and we encountered issues. Specifically, when we calculated the integrand of the KLD, it resulted in dividing by zero. In the particular case where the integrand was 0/0, we could resolve the issue. We can define the indeterminate form 0/0 as zero since this corresponds to the situation where the opinions agree. We couldn't resolve the issue when the integrand was (1/2ε)/0 because the integrand would be infinite, implying the KLD is infinite, and we can set no sensible value for the integrand. We surmised that the KLD is untenable for the HK bounded confidence model. Hence, we now discuss this limitation in the discussion section.

* Fig.1: The resolution of the figure was quite low on my computer. Consider using a better resolution in the final version.

Response: We have included too much in Fig 1 making the figure too low resolution. We have separated Fig 1 into 2 figures to improve quality of images.

* Exponential expansion: Isn't the exponential expansion damning of the KLD measure? Visually the opinions have clustered (Fig. 1), and only the lack of variance leads to many opinions. This leads to several counter-intuitive results (see also the second "phase" of Fig3 and 5).

Response: Exactly, but we noticed that the exponential expansion rate was related to the number of clusters. We know that the Martins model never reaches full consensus and only gets arbitrarily close. The Martins model has this behaviour because uncertainty halves when an agent interacts with another agent they agree with. KLD is calculated using an agent's uncertainty and hence would be affected by the geometric decay of uncertainty. Knowing the mechanics of the Martins model, we devised a method to estimate the cluster count from the exponential expansion of the mean KLD.

* p14: I don't think that interpreting the spectral radius as the size of the component is correct. It also captures connectivity to some extent. For example, the radius will slowly transition from ~n/2 to n-1 if we start with two disconnected, fully connected blocks and slowly connect them until we have a single block where all connections exist.

Response: We agree with the reviewer on this point as it’s unclear what spectral radius means when “two disconnected, fully connected blocks” begin to connect. We believe that the spectral radius is the effective largest cluster size. A spectral radius of ρ tells us that a graph of size n is equivalent in connectivity to a fully connected graph of size ρ.

* The way the network structure is defined is a missed opportunity, in my opinion. The graph structure is wholly derived from the opinion values themselves. Hence it does not add a second layer of data (e.g., a social network on which opinions are exchanged). In the end, the proposed graph measures could largely be computed without references to the graph (e.g., for the min-cut and the HK model: slide a line across the opinion line and find the location such that there is the least agents within a fixed band. Report the number of agents as the size of the cut.)

Response: We have implemented the method suggested by the reviewer and found some interesting results. The method initially matched the min-max flow methods results for the HK bounded, but when the simulation entered polarisation, instead of reaching zero like the min-max flow rate, what the review suggested produced the size of the smallest cluster. Due to this novel result lying outside the scope of this manuscript, we have included a new appendix (S3 Appendix), which discusses what we have found.

Reviewer #2:

1. The main innovation of this manuscript is limited and unclear. The reviewer suggests that the authors spend a great amount of effort to better clarify the novelty of their proposed technique. In overall, this manuscript lacks accurate discussions on the differentials of the proposed method. Why is this proposed technique better than others? What exactly are the contributions of the proposed technique? The reviewer suggests the following approaches to solve this issue: comparisons could be made by implementing and running other techniques over the same raw data. Such a comparison can be made in terms of efficacy or efficiency (how good the proposed technique is in using the available resources - i.e. time, computation...).

Response: Our goal with the manuscript was to develop more compelling methods to measure polarisation, i.e. to improve our understanding of polarisation. Although computation time and efficiency are important, it is not the focus, with the focus being whether these methods can measure how much agreement there is in society more precisely. But comparing the new and old methods would improve the manuscript, so we have expanded the beginning of the results section to include the results of running cluster counting algorithms and the ‘y-statistic’ on the simulated data.

2. Some figures, especially figure #1, shown in this manuscript look blurred; therefore, the authors are suggested to improve their quality and legibility.

Response: As said to reviewer 1, we have included too much in Fig 1 making the figure too low resolution. We have separated Fig 1 into 2 figures to improve quality of images.

3. In the abstract, it would be better to highlight and specifically stress what has been done in this research work. The present abstract fails to accurately reflect the novelty of the proposed framework.

Response: We have updated the abstract to discuss more about the results of the paper.

4. In the introduction section, many recent and relevant contributions close to the authors’ research work are not reviewed. The authors should consider and mention some more recent and relevant references to the literature review to create a proper perspective of the existing state-of-the-art in the area of interest.

Response: We have done more literature review and included mention of more state-of-the-art methods of polarisation in the introduction.

5. More details and references concerning Kullback–Leibler divergence and the Hellinger distance metrics are required. Moreover, the paper shows a lack of references. Some equations and formulations in the text need references and/or mathematical demonstrations.

Response: We have now included a more detailed literature review of KLD and H distance and provided more mathematic derivations of equations within the supporting information as appendices S4 and S5.

6. There are some English typos throughout the manuscript (especially, in the abstract, introduction, and conclusion). The authors are needed to thoroughly revise the paper and correct the English grammar mistakes.

Response: We have thoroughly proofread through the entire manuscript paying close attention to the abstract introduction and conclusion section to remove all typos in the manuscript.

---

## [Decision Letter · Decision Letter 1]

13 Sep 2022

Mathematical measures of societal polarisation

PONE-D-22-04726R1

Dear Dr. Adams,

We’re pleased to inform you that your manuscript has been judged scientifically suitable for publication and will be formally accepted for publication once it meets all outstanding technical requirements.

Kind regards,

Ismael González Yero

Academic Editor

PLOS ONE

Reviewers' comments:

Reviewer's Responses to Questions

**Comments to the Author**

Reviewer #1: All comments have been addressed

Reviewer #2: All comments have been addressed

2. Is the manuscript technically sound, and do the data support the conclusions?

Reviewer #1: Yes

Reviewer #2: Partly

3. Has the statistical analysis been performed appropriately and rigorously? 

Reviewer #1: N/A

Reviewer #2: Yes

4. Have the authors made all data underlying the findings in their manuscript fully available?

Reviewer #1: Yes

Reviewer #2: (No Response)

5. Is the manuscript presented in an intelligible fashion and written in standard English?

Reviewer #1: Yes

Reviewer #2: Yes

6. Review Comments to the Author

Reviewer #1: All the comments I have raised have been addressed in this new submitted version of the manuscript .

Reviewer #2: The authors have improved the quality of the paper in terms of organization and writing. In addition, they have properly addressed my comments. In the reviewer’s opinion, the current manuscript can be accepted for publication in the journal of Plos One.

7. PLOS authors have the option to publish the peer review history of their article (what does this mean?). If published, this will include your full peer review and any attached files.

Reviewer #1: No

Reviewer #2: No

---

## [Editor Report · Acceptance letter]

26 Sep 2022

PONE-D-22-04726R1 

Mathematical measures of societal polarisation 

Dear Dr. Adams:

I'm pleased to inform you that your manuscript has been deemed suitable for publication in PLOS ONE. Congratulations! Your manuscript is now with our production department. 

Kind regards, 

on behalf of

Dr. Ismael González Yero 

Academic Editor

PLOS ONE